# Consumption Culture and Critical Sustainability Discourses: Voices from the Global South

**Arindam Das** [1,*] **and Pia A. Albinsson** [2]

[1] Department of Language & Literature, Alliance School of Liberal Arts, Alliance University, Bengaluru 562106, India
[2] Department of Marketing & Supply Chain Management, Walker College of Business, Appalachian State University, Boone, NC 28608, USA
[*] Correspondence: arindam.das@alliance.edu.in

**Abstract:** Our qualitative critical research intends to examine the meta-normative features of the sustainability discourse of the marginalized Global South through sociological critique of (neo)colonial and anti-sustainable consumption. Using a critical lens, we discuss two community performances of sustainable consumer culture from the Global South to highlight the subversive consumption performances in the Global South market, which has the potency to ontologically denaturalize the Global North market's standard-normalized Western discourses of sustainability that tend to legitimize social inequalities and the seizing of agency by marginalized consumers of subsistence marketplace. The article contributes to both sustainability and consumer culture literature by proposing a new research agenda: the way sustainable consumption culture projects and negotiates identity in the Global South, especially at the margin. We highlight how traditional sustainable prosumption of subaltern subjects of the Global South resists power practices promulgated by Western capitalism, neoliberalism, and neocolonization.

**Keywords:** sustainability; consumption culture; critical analysis; neocolonial; SDGs; global south; indigenous





## 1. Introduction

Sustainability research in business draws largely from the discussions regarding depleting world resources, anthropocentric negligence, and re-improved structures that engender ethical and conscious decisions (when sustainability is integrated into decision-making processes, organizations are more likely to consider social and environmental impacts [1]). The agenda toward sustainability research primarily focuses on initiatives seeking to minimize the materialistic impact on the environment and provide solutions to societal challenges for a holistic human well-being, socio-cultural equity, and nature-positive existence. Tackling macro issues of climate change, nature challenges, detrimental technological omnipresence, absence of intra-generational, and multi-societal equity, sustainability research positively impacts sustainable development (SD). SD aims to dissolve disparities among communities across regions and groups, and develop policy change (at governmental and/or organizational levels). Such changes help recreate a society from a perspective of equity, that rules out disparity and promotes better quality of life and well-being [1]. However, the primary concern of all SD revolves around the issue of 'resources', i.e., the capacity misuse, depletion, politicized allocation disparities, and uncertainties in management methodologies. The competitive strategization of corporate social responsibilities (CSRs), formed to address environmental and natural issues, are addressed by the resource-based views (RBV) of sustainability [2]. Corporations' investment in socially responsible activities has consequences in the creation of intangible resources that impacts corporate image [3] and 'attractiveness' [4] with stakeholders. The enhancement in corporate accountability

through CSR consolidates further when social responsibilities particularly become sustainable responsibilities. Additionally, sustainable social responsibility tries to penetrate 'bottom-of-the-pyramid' markets through legal, moral, or economic means [5–7]. Nevertheless, the RBV of sustainability, adopted by neoliberal corporations, gets highlighted over any other expressions of sustainability, which leads us to reconsider critical perspectives of sustainability, an area with meagre research. Not only quantitative and positivist research on sustainability, but more critically qualitative work is required to probe the prevalent issues and narratives of sustainability. Critical research probes the socio-political–economic dynamics between the 'have' and the 'have-not' groups in society. With an objective to seek and represent social justice, qualitative critical research focuses on the power asymmetry between the center and the margin in the neoliberal market structure. Critical sustainability research seeks to uncover the ways in which power is exercised and maintained in society, and works to empower marginalized groups by giving voice to their perspectives and experiences of traditional sustainability methods. Problematizing and decolonizing the anti-sustainable rhetoric of the Western-globalized consumer culture through politicized consumption behaviors [8,9], the marginalized consumers of the Global South resist being faded into oblivion or being appropriated by the monologic global Western capitalism, and seizes agency for self-representation. A lack of thorough research in this area is our primary motivation behind this work. Our research intends to critically project the meta-normative characteristics of the sustainability discourse of the marginalized Global South through sociological critique of (neo)colonial and anti-sustainable consumption. We highlight the subversive consumption performances in the Global South market that have the potency to 'ontologically denaturalize' [10] (p. 10) the Global North market's standard-normalized Western discourses of sustainability that tend to legitimize social inequalities and the seizing of agency by marginalized consumers of subsistence marketplaces.

One primary area in sustainability research that remains to be thoroughly examined through a critical lens is the interconnections between markets, consumption practices, and sustainability discourses from the Global South. Critical sustainability studies are not altogether new [11,12], and critics and authors of environmental studies have discoursed the demand for social justice for a sustainable future over the capitalocene and its exploitative agenda [13]. Most of the studies in critical sustainability by Anthropocene scholars critique how paternalistic sustainability programs, a further extension of the institutionalized Western model of development and neoliberalism, fail to address the basic requirements of the poor and marginalized of the world, which should otherwise have been a priority [14]. Focus areas such as equity, justice, well-being, and recognition of marginal eco-system have been the primary agenda of critical sustainability research [15]. Nonetheless, understanding the possibilities of an ecologically sustainable future through the consumer culture of a marginal community was never attempted. Therefore, the sustainable practice, narratives, performances, and discourses of the subalterns of the Global South that indicate their identity consumption is the focus area of this research. The articulations of sustainable consumption that are conceivable at the local micro sites have been overridden by the grand discourses of capitalistic consumption, managed by the global neoliberal markets. Responding to the Special Issue's call for papers on the 'Consumption culture and Sustainability Discourses', we acknowledge the high anthropocentric consumption pattern across the globe. However, the more sustainable, alternate consumption practices are also noticed at the level of identity negotiations [16], socio-cultural struggle [17], bottom-of-the-pyramid market performances [18], resolution dialogues [19], issues of race, multiculturalism, transnationalism [20], performances of nation-making or nationalism [21,22], environmental politics [23], human rights and justice for minority community [24], traditional ethics and aesthetics [25], gender interpretations [26], subaltern voicing [27], in addition to memory and culture [28]. Deconstructing the unsustainable ideologies of the Global North, the everydayness of consumption culture [8,9] of the Global South is an indication of an emerging counter discourse that dismisses capitalism. The meta-normative characteristics of the sustainability discourses from the Global South may

be substantiated through the critical sociological theories of consumption. This article will analyze and discuss two community performances of sustainable consumer culture from the Global South through a critical lens. The findings contribute to both sustainability and consumer culture literature by ushering a hitherto never-discussed research agenda: how sustainable consumption culture projects and negotiates identity in the Global South, especially at the margin. We highlight how traditional sustainable prosumption of subaltern subjects of the Global South resist power practices promulgated by Western capitalism, neoliberalism, and neocolonization. However, before that, we will highlight the need for critical sustainability studies through a review of literature.

## 2. The Neocolonial Anti-Sustainable Market Rhetoric, and the Need for Critical Sustainability Discourse

The biologist Eugene Stoermer first used the term 'Anthropocene' during the 1980s, and afterward an atmospheric chemist, Paul Crutzen, promoted it more comprehensively in the mid-2000s [29]. These two researchers characterize the 'Anthropocene' as another geological age, succeeding the Holocene (one that had a softer human impact on earth), where people have arisen as the focal geographical power, molding the worldwide environment and climate [29]. Although the date for the beginning of high Anthropocene varies from scholar to scholar, for our research, we will abide by Lewis and Maslin's [30] and a host of other votaries according to whom the high Anthropocene begin with European colonization. With colonization and the consequent spread of modernity and capitalization, the man-centric anthropogenic consumption increased, thus bringing environmental destruction and anti-sustainability in its wake. Such anthropogenic colonization, impacting sustainability, was not merely at the level of 'transplantation of the flora and fauna of 'Home' (Britain) to the 'new country' (viz. Australia, New Zealand) by settlers' with the vain hope to 'improve the colonies' and suit the needs of the colonial man [31], but also, as evident, the colonial man's greed for natural minerals (i.e., diamonds and ivory in Africa; gold and coal in India, Australia, and New Zealand). Indeed, such attempts to manipulate and harness natural resources to self-centric need generated 'eco-anxiety.' Colonized/marginal subjects suffer from anxiety about environmental calamity, just as they are no less helpless against the xenophobic, racist, and fundamentalist ways of the perpetrators of environmental degradation. However, the impact was more than the generation of "eco-anxiety". The consumerist agendas of capitalist colonial modernization and hegemony over colonized consumers' psyche in integrating conspicuous consumption [32] led to the commodification and marketization. This accentuated anthropogenic consumption. Hence, anthropogenic colonial consumption was also an outcome of power relation between the 'developed' West and the 'underdeveloped' rest. In this production of power relations—between the modern colonial marketeer and the colonized consumers—a 'progressive' Western economy was postulated across the colonized world. New domains of life, colonized through capitalist advocates of modernity [33], made marketization rampant. As it seemed, colonization was interchangeable with marketization [34]. The market logic was made omnipresent in colonized space and psyche.

The non-human world of colonized spaces was perceived as a 'standing reserve' for state extraction to consolidate its capitalist motives [35]. The incommensurable relationship of the power hierarchy between the colonizer and the colonial land accentuated anthropocentric ideology. For example, due to the 'coal colonialism' (a product, in a commercial sense, unknown in pre-colonial India) in India [36], the Adivasi (Aboriginal Indians from various underdeveloped regions of India) displacement from traditional land (the state of Jharkhand) was imperative. A massive loss of flora and fauna was also noted: 'When coal mines destroy and degrade forested tracts, devastating the local flora and fauna—and along with them the lives of local poor—then one begins to see through the politics of selective dispossession that hides within the official messages of development' [37] (p. 145). Theorizing extortionist moves such as 'ecological imperialism', Alfred Crosby reminisces a poem from Walt Whitman that talks about the booty bounty from the colonized land:

'Land of coal and iron! land of gold! land of cotton, sugar, rice!/Land of wheat, beef, pork! land of wood and hemp! Land of the apple and the grape!/Land of the pastoral plains, the grass-fields of the world! land of those sweet-air'd interminable plateaus!/Land of the herd, the garden, the healthy house of adobie!' (Walt Whitman, 'Starting from Paumanok', as cited in [38] (p. 294)).

The ideals of 'ecological imperialism' that led to unsustainable consumption resulted in an accumulation of wealth. To expand the capitalist economy, yielding a surplus of demand, colonized subjects needed to be hegemonically conditioned. The case of alcohol consumption by the Australian Aborigine is a case in point. Even before the arrival of the whites, the Aborigines of Australia consumed mild alcohol made of fruits, flowers, leaves, stems, and honey. However, the community prohibited excessive consumption, and large/mass-scale (industrialized) production, storage, and consumption were absent [39]. After the arrival of the colonizers in Australia, there was a mass-scale alcohol consumption among the Aborigines, causing higher mortality rates, domestic violence, road accidents, mental health problems, suicide, and aggressive behavior [40].

Such colonial 'modern' practices of unsustainable consumption are also witnessed in the neocolonial/neoliberal market performances. The Global-North-influenced unsustainable, anthropocentric consumption systems violate the human rights and democratic values of the subalterns. Profit-centric ideology of the large and concentrated transnational corporations, financial institutions of neocolony, exploitative bourgeoisie interests, and the politico-economic system of the nation state that institutionalize and reinforce socio-economic disparities are the key actors and successful perpetrators in mis-responding to sustainability discourses and frameworks. For example, the Brazilian government and its policies have renewed concerns about an unsustainable economic development [41]. With an increase in deforestation and resource extraction (construction of a hydropower station in the Trombetas River) in the Amazon basin, climate alteration is being felt regionally. Infectious diseases have spread out even to neighboring countries: 'Forest degradation in the Amazon has facilitated the spread of diseases with potentially large social and economic impacts, both locally and globally. Multiple pathogens thrive under land-use changes, deforestation, and poverty, causing a significant burden on the health and economic prosperity of Amazonians. The health dimension is scarcely included in discussions around development of the region' [41] (p. 2). Further, with an impact on biodiversity, fishes, and animals, resources for the sustainable living of the poor people have depleted. Indeed, such unsustainable practices in the Amazon basin have affected the cash-poor Amazon riverine households of Brazil:

'Hunger is intensifying due to many interacting factors: as COVID-19 assails the Amazon; Jair Bolsonaro's government shreds once effective social welfare programs and cheerleads forest destruction; and climate change-driven extreme floods and droughts are on the upswing. Experts warn that these circumstances intensify threats to food security for the poor in remote riverine communities' [42] (n.pg).

Even the UN Sustainable Development Goals (SDGs) and its associated agenda of progress, development, and economic growth have been critiqued on the grounds of promoting neoliberal aspirations through the facilitation of privatization and marketization [43]. The SDG document promoting Public Private Partnership (PPP) calls 'for worldwide action among governments, business and civil society to end poverty and create a life of dignity and opportunity for all, within the boundaries of the planet' [44]. Although there is a strong focus on the pivotal role of the nation state, capitalist corporates, and empowered and agency-wielding civil society meant to generate a positively 'enabling' environment for the poor, the UN document fails to involve the voice/space of the poor and the subaltern. The paternalistic and benevolent communication of the development narratives insinuate inequitable arrangements. These are not only attempts toward the erasure of subalterns, but also the steeped-in language of neo-colonizers [43].

*Sustainable Consumption as Identity Politics at the Margin*

'The meanings of an identity are, in part, the products of the particular opportunities and demand characteristics of the social situation, and are based on the similarities and differences of a role with related, complementary, or counter-roles' [45] (p. 84). This statement presents a way of life as generally context-specific, relational, and thus dynamically evolving. As the setting, and the social relation with it (or with the subjects in that setting), changes, so does our identity. Identity is impacted by a variety of factors: cultural, social, economic, ethnic, individual factors, community relationship/interactions. Further, identity is a matter of constant flux, not absolute fixity. It changes and evolves through multifaceted experiences that a subject is exposed to over a length of time. In terms of the market, the identity of a subject depends on consumption relationships between the subject and the services or products. When the subjects consciously consume, they identity-perform [46]. Hence, our identity is highly prone to be influenced and designed by the market. Moreover, with 'the interaction between cultures and market (are) accelerating in the global economy' [47] (p. 249), our identities tend to be conditioned by neoliberal strategies. However, it would be a gross generalization to say that all consumers tend to acculturate the neoliberal ideologies as conveyed through goods and services available in the global market [48]. Acculturations in the market are multifaceted phenomena. If capitalism and global market discourses accentuate a standardized, non-heterogenous, global consumption, then the 'differentiating impact of globalization strengthens or reactivates national, ethnic, and communal identities; and the pattern of interrelationships fuels a hybridization of social life' [49] (p. 65). The negotiated, mediated identity in the global market straddles between its traditional and contemporary condition [50]. Such localized, nativist consumption may also be consciously sustainable and subversive of the neocolonial market agendas. Imbued in the politics of identity, sustainable consumption, especially those consuming at the margin of the Global South, can redefine their identity (both individual and collective): socio-cultural, economic, racial, ethnic, general well-being, and relating gender. The sustainability-defined relation of the subjects to their consumption objects leads to the negotiation of identity in the global market. A few examples may further illustrate our point. The Australian Indigenous understanding of sustainable consumption may be witnessed in the Wangan and Jagalingou people's protest against the Adani Carmichael coal mine project. These fourth-world people strengthen their ethnic and nationalist identity consumption (where environment is consumed as an extension of self and an ancestral mythical essence) and counter the neocolonial/capitalist market through environmental activism [51]. Similarly, in search for a post-carbon pre-colonial environmental ontology, the South African indigenous women narrate an identity that is framed as feminist–socialist–environmentalist concerns [52].

In establishing a relation between sustainable consumption and identity of the marginalized, there are scopes for political negotiations and resistances. Seeking agency by disagreeing to be passive recipients of neoliberal and neocolonial Westernized episteme of 'development', the subalterns may consume to their choice. Such consumption choices may be strategically opposed to that of the neocolonizers, and hence sustainable, as framed against unsustainable. The acknowledgement of such alternate sustainable consumptions of the marginals accentuates democracy. According to Sen [53] (p. 155), for human development in a democracy, it is necessary to give prominence to free choice. This helps one live the life one values: 'the people have to be seen, in this perspective, as being actively involved—given the opportunity—in shaping their own destiny, and not just as passive recipients of the fruits of cunning development programs' [53] (p. 53). Self-categorization and identification are ongoing dynamic projects highlighting a subject's epistemic choice-based existence to a privateness or to a community. Sustainable consumption, whether at the individual level or at the community level, is a marginalized ontological existence digressing from standardized practices. Additionally, this makes it a politicized identity choice, and the demand for its recognition a democratic affair. In the following section, we

highlight two community performance narratives from the Global South to illustrate how marginal communities negotiate identity by making sustainable consumption choices.

### 3. Research Objectives and Research Questions

Through sociological critique of neocolonial and anti-sustainable consumption, this research, which bridges consumer culture theory and social sustainability research, aims to delve into the meta-normative characteristics of the sustainability rhetoric of the subalterns of the Global South. We intend to highlight the subversive consumption narratives in the Global South market that have the potential to ontologically denaturalize the Global North market's conventionally normalized Western discourses of sustainability and social inequalities. We seek to understand how the marginalized consumer culture helps seize agency in the subsistence marketplace and recreate a new sustainability paradigm. We discuss two community performances (i. *Bondas* of Odisha, India; ii. *Patachitrakars* of West Bengal, India) of sustainable consumer culture of the marginalized communities of India.

We ask two pertinent research questions relevant to both subaltern communities:

a.   How does the traditional consumer culture evoke a counter discourse of sustainability, beyond the normalized globalized version of the same?

b.   How does the traditional sustainable consumption culture and its everydayness of politics help build agency and community identity?

We probe to understand the psyche of the subjects through their identity consumption and responses to traditional wisdom around perspectives of sustainability. We analyze the discourse of both the conscious and unconscious cultural consumption performances of the two communities.

### 4. Research Method

To study the sustainability narratives and discourse used by the two marginal communities in rural India, we focused on multiple sources of qualitative data, which we analyzed through a critical lens. Engaging with data with a critical attitude means that researchers 'scrutinize previous theories and research of others to detect the presence of mistakes, oversights, incompleteness, inadequacies, holes in the argument, fallacies, contradictions, vagueness inadequate data' [54] (p. 313). We analyzed publicly available archival data, such as published reports, digitally archived materials (songs, art posters, art narratives), and online interviews. Therefore, communities that engage in similar sustainability practices, but without public documentation, were excluded from the selection process. The communities chosen for our study were two marginal communities from different regions of India, and differed in demographic conditions, such as size, ethnicity, language, employment, and source of income. Such a varied choice was intended to make the approach more holistic. The two communities chosen were the *Bondas* from Odisha and *Patuas* from West Bengal. Our analysis utilized a thematic and hermeneutic approach common for ethnographic studies using narrative and text analysis, triangulating various emergent themes [55–58]. Specifically, we systematically analyzed documented texts, visual, and audio–visual data, available in research reports and social media. We identified the research questions (as mentioned previously) and conducted thematic analysis to classify the content of the text. This provided insight into cultural, historical, social, and economic conditions of the community in which these texts are presumed.

### 5. Narrative Performances

*5.1. Bondas and Sustainable Farming*

The first community we chose to study was the *Bonda* tribe and their sustainable farming practices. The *Bondas* lives in the Malkangiri district, Odisha, India, which is located south–west of Kolkata. *Bondas* are a vulnerable Austro-Asiatic tribal group residing in the hilly forest areas (there were 12,321 *Bondas* according to the 2011 India Government Census; one of 75 classified groups referred to as particularly vulnerable tribal groups [PVTG] in the country). Although a Government of India economic report claims the spread

of 'multilingual' education (mother tongue and English) and consequent socio-economic progress among the *Bondas* [59], the reality is far from it. The implementation of tribal upliftment programs by government agencies, such as the Bonda Development Authority (BDA), are not very successful, owing to the general non-acceptance of outsiders by the tribals [60]. The primary livelihood and economic assets of the *Bondas* are from forest gathering, hunting, and cultivating on hill slopes (i.e., hillocks) [61]. The tribe sustains itself by consuming traditional food crops, collecting food from the mountain, and brewing traditional liquor [62]. An ethnobotanical analysis of the available variety of plants and herbs consumed by the tribe as medicine is indicative of the sustainable ecosystem operated by the *Bondas* [63]. Although globalization has forced them to move away from traditional millet-centered mixed-crop systems to more non-traditional commercialized crops such as paddy (i.e., rice), which generates a better yield, the *Bonda* women are determined to fight back and resist. '*Bonda* women (are) reverting to the cultivation of native millet varieties—finger (*ragi*), foxtail (*kakum* or *kangni*), barnyard (*sanwa*), proso (*chena*) and pearl (*bajra*) millets—which are climate-resilient and ensure the community's food and nutritional security' [62]. Figure 1 below [64] briefly mentions the cultivation method on the hillock adopted by the *Bondas* and resultant positive impact on sustainability. Such methods are centuries old and are indicative of traditional community wisdom. Learning to counter the heavy rainfall, flash floods, and landslides, traditional cultivation is carried out on the hillocks. Moreover, traditional crops requiring less rainfall, such as millets, are grown. Such crops may have lesser commercial value, but are sustainable enough for community consumption. There are various advantages of *dangar* or hillock cultivation of millets carried out by the women of the tribe. Millet requires less water and, unlike rice cultivation on flat lands, does not retain water that produces methane. Less generation of methene gas due to lower temperature decreases pollution. The cultivation site becomes a breeding ground for local fauna (birds, insects, butterflies, etc.). It has been witnessed that the temperature during summer (May–June) decreases to 35 °C on the hills of Malkargiri (while temperature in other areas of the district increases to 45 °C) post-continuation with the cultivation of millets. This is indicative of global warming management at the micro level [64].

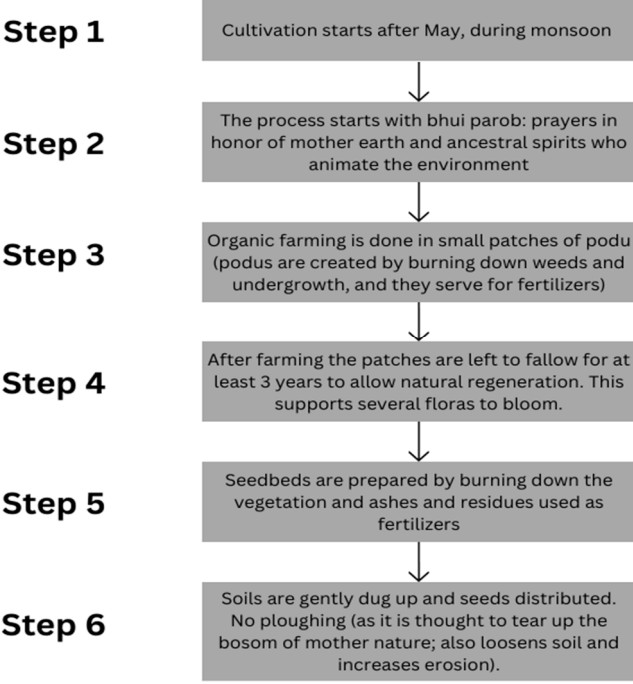

**Figure 1.** *Bonda* tribe's *dangar* (Hillock) cultivation method.

Agriculture and farming are intrinsic to the myths and religious practices of the *Bondas*. *Bondas* celebrates the *Patakhanda* (the holy sword) festival with great grandeur on the Monday that comes right before *Magha Purnima* (the full moon day of Magha). The enormous banyan tree's circumference is more than fifty meters (a particular banyan tree at Mudulipada being the epicenter of a *Bonda* religious festival), around which the *Patakhanda* is secretly hidden. Its branches and leaves cover at least 2000 square feet of space. A platform of stone decorations around the tree marks its importance. The tree with its lush branches offers a cool, shaded area, warding off intense heat. Tribal discourse claims that despite the intense heat during certain seasons, no tree near this sacred location ever perishes or dries up. The principal priest who does the *Patakhanda* worship at the banyan tree is called *Sisa*. He is also known as *Manadora* (*Manadhara*) because he transports the *mana* (a little bamboo container filled with paddy or any other grain said to be the goddess *Lakshmi* or *Bhudevi*, the Goddess of wealth and earth) during the puja. The *Sisa* sanctifies the *Sila*, *Silapua*, and *Chaki* (the grinding stones) with the oil before bathing them in water brought from a nearby holy waterhole in a gourd container. The rice for the puja is then prepared by the priest using the oblation fire, or *charu anna*. *Sisa* pays his obeisance to the banyan tree before ascending it to collect the secret sword. Ducks, goats, lambs, and hens are sacrificed with the holy sword and cleansed with water from the nearby waterhole. *Mantras* (hymns) are chanted to the Lord for a plentiful harvest: 'Sumu sarai, pakna gaja Demta!/Maprunan satare manek/Na duiman denata' (translated: 'With your help, the crop will multiply into two units from one') [65] (p. 69). The agriculture God, integral to the part of the *Bonda* life, mythically oversees sustainable farming, with absolute control over anthropocentric exploitation of nature.

Among the *Bondas*, agriculture and farming are mostly carried out by women [66]. This is a positive feminization of agriculture, and the *Bonda* women's resistance to chemicals, fertilizer, 'modern'/'scientific' methods of farming, protection of crops, and storages bring tribal knowledge, environment, and sustainability together in a single frame [67]. Countering the developmental praxis of modernity that removes the tribals' rights and displaces them from their land, *Bonda* women are countering climate change through indigenous methods effectively. Specifically, through the performances of their intrinsic tribal knowledge systems (the skills and philosophies developed by a society through their long historical interaction with the immediate nature and environment), the *Bonda* women are deploying resistive cultural/identity politics to hegemonic systems of neoliberalization. For example, the *Bonda* women do not use chemical fertilizers and allow for birds to eat insects, limiting potential harm to crops. Instead, they use natural pesticides made from *neem* (Azadirachta indica) trees. They also involve their traditional knowledge in seed selection and protection. Such microlevel traditional farming practices are not only sustainably strategic but protect and help strengthen the identity of the tribal women. This is a true ecofeminist way to consolidate the *Bonda* women's identity in the face of neoliberalization.

Within an Indian context, Adivasis or tribals are those whose 'social, cultural and economic conditions distinguish them from other sections of the national society, and whose status is regulated wholly or partially by their own customs or traditions or by special laws or regulations' [68] (p. 21). Even the consumption habit of the Adivasis, such as *Bondas*, are intrinsically different from others. Unlike the 'mainstream' exploitative neocolonial consumption habit impacted by modernization, the Adivasis of India consume in limits. Rather than exploiting the land, which is a part of their identity and existence, the Adivasis cultivate with the purpose of localized consumption and embrace biodiversity as conservationists. The 'anti-modern' consumption habit of Adivasi minorities of India strengthens their culture, identity, history, and existence. It acts as a politicized everydayness against any dominant exploitative paradigm.

The Adivasi song, 'Gaon Chhodab Nahi', inspired by Bhagwan Maajhi, leader of the Adivasi struggle against bauxite mining in Kashipur, Odisha, brings forth the resistance of the Adivasis to the pressures of modernization and unsustainable consumption that intends to displace the tribes and appropriate their culture and heritage:

'We will not leave our village!

Nor our forests!

Nor our mother-earth!

We will not give up our fight!

They build lands, drowned villages, and built factories

They cut down forests, dug out mines, and built sanctuaries.

. . .

Oh God of Development pray tell us, how to save our lives?

. . .

You may drink your colas and bottled Bisleries

How shall we quench our thirst with such polluted water?

Were our ancestors fools that they conserved the forests?

Made the land so green, made rivers flow like honey?

Your greed has charred the land and looted its greenery!

. . .

We will not give up our fight!'

Consumer resistance literature and its varied manifestations highlight resistive-negotiatory identity projects at various levels. There have been substantial attempts at theorizing consumer resistance in the last two decades. Such resistance theories have rethought consumer agency through resistive-reformist to radical consumer movements, individual to collective counter-narratives in the market, consumer forays to alternate market mix, and even moving beyond marketing institution or using a non-marketing institution [69]. Adding to this, Varman and Belk [70] talk about the boycott of the hegemonic market and the creation of an alternate consumptionscape. Anti-commercial consumer rebellion against institutionalized marketing [71] or denouncing exploitative (even potential) consumption [72] was discussed in consumer resistance literature. Consumer culture theorists and marketing academics had often perceived consumer resistances as consumer agency enablers, especially for the market-marginalized consumers [5,73–75]. From the perspective of sustainable consumption practices as resistance to neocolonial excess, consumer theorists have upheld the responsibilities of green activists or environmental activists [35,76–79]. There have also been critical studies of the limitations of consumer resistance, with potencies of itself becoming a hegemonic grand discourse [80]. However, consumer literature fails to adequately address the subaltern issues from the Global South, especially with reference to sustainable consumption as a resistance practice.

A hermeneutic approach to the consumer culture of the marginalized Global South population would 'focus(es) on the symbolic meanings and processes by which individuals construct a coherent sense of self-identity' [81] (p. 389). The individual and community narratives of sustainability and pro-nature consumption (as in the above Adivasi song) manifest a desire for freedom of choice, existential rights, and a politicized demand for anchoring and buttressing identity [82].

*5.2. Patachitrakars, Sustainable Counter Episteme of Traditional Aesthetics, and Rethinking Modernity*

In this section, we discuss our analysis of the *patuas* or *patachitra* artisans or *chitrakars* from a small village called Naya in West Bengal (130 km south–west of Kolkata and house to about 250 *patachitra* artisans), India, and how their art practices help promote SDG goals and subvert neocolonial platforms of transaction. *Chitrakars* are mostly poor Sunni Muslim artisans who engage in scroll painting (*pata*) (now considered as intangible cultural heritage of ICH by UNESCO, and earning a Geographical Indication (GI) tag—"Bengal Patachitra"—in 2018), depicting elements from Hindu mythology and even narrating songs while unscrolling the scrolls. According to *patua* Gurupada Chitrakar, there are almost

80 families and about 275 *citrakars* currently in Naya [83]. With their bags of scrolls, the *Patuas* travel to villages and visit from house to house. They tell tales as they unfold the scrolls in exchange for payment, cash or kindness. There is a recent consumer interest and media focus on this visual storytelling community. Several NGOs work toward the community's socio-cultural upliftment and showcase their products in the markets, the annual artisans' fair-cum-exhibition conducted by the Government of West Bengal at district levels, and various international cultural exchanges (Europe, South Asia, USA). Owing to a rise in market demand, the product diversification of *patachitra* has taken place. The *patuas* receive orders for dress materials and home décors from boutique shops and for the decoration of *pandals* (a temporary decorative structure set up to venerate deities) during the biggest Hindu religious festival in West Bengal, the *Durga Puja*. However, all these exposures and efforts are not enough, and the community of singing painters continues to remain economically and socially downtrodden. It is worthwhile to note that the centuries-old *patachitra* tradition (2500 years old see [84]) uses sustainable natural products for their paintings. The very word 'pata' comes from the Sanskrit word 'patta' or a 'piece of cloth'. The bold, bright, multi-colored paintings are composed of natural dyes (made from trees, flowers, and minerals). A recycled soft fabric is pasted on the back of the scroll to make it stronger. Gum from wood apple (*Limonia acidissima*) is mixed with the colors to give it permanence, and finally, it is sundried to darken [85].

However, our focus is not the sustainable art form, per se, but how the sustainable art form is employed to address, albeit unknowingly, some SGD goals: good health and well-being (SDG 3); quality education by increasing vaccination awareness (SDG 4); gender equity (SGD 5); clean water and sanitation (SDG 6); climate action (SDG 13); peace, justice, and strong institution (SGD 16) [44]. We note the way artisans strategically politicize and subvert modern neocolonial transactor zones or media of art expressions, thereby deconstructing technology, social media, and the marketization sallies. Beyond the consumerist agenda, the *chitrakars* expand their cultural product in the market with a non-profit motive, addressing sustainable social causes (generally and particularly observed during COVID-19). The *chitrakars* defamiliarize their art form, which could otherwise supply the fetish consumption and be appropriated/canonized/commoditized by the urban art market. The *chitrakars* reproduce their *patas*, from time to time, addressing social issues. Such strategization is helpful to prevent their art form being prey to neoliberalist capitalist market structure and a method to show how sustainable consumption is possible through marginalized aesthetics.

The COVID-19 pandemic has impacted the art ecosystem of the *patachitra* artisans of Naya, West Bengal. Patronages ceased, orders got cancelled (hugely reduced due to lower buying power of customers), fairs and exhibitions closed, performances stopped, travelling was banned, the art space became digital, and all these significantly reduced *patuas'* income. During COVID-19 lockdowns, to sustainably keep their artform vibrant, with assistance of various NGOs, the *patuas* rethought from mythic, religious, and traditional themes of *patachitra* to more social and contemporary ones, and moved to surrounding villages to spread awareness. Having previously addressed themes such as proper sanitation, maternity and child health, HIV-awareness, right to education, the community moved toward more socially demanding issues of the COVID pandemic [86]. Commenting about songs such as these, Korom [87] observes:

> '[ . . . ] they needed to innovate for their performances to remain fresh and relevant to contemporary society and the issues that confronted it. The *Patuas* thus developed a new genre of a song called samajik gaan, or "social song", which did not replace pauranik gaan, but supplemented it'.

In his seminal book on the *patachitra* artisans of *Bengal*, Korom highlights how the singer–painters of Bengal have self-fashioned a modernity of their own through their intrinsic sense of resilience and ability to negotiate modernity through indigeneity. Indeed, the precarity that has underpinned the *Patua* existence for centuries due to oppressive penury, instability, and marginalization is compensated by their resourcefulness, innova-

tiveness, and resilience that has ensured the continuity of their practice for centuries [88], as referenced in [86]. However, the above authors [86–88] fall short of seeing this cultural-heritage narrative as politicized and strategic defamiliarization to tackle identity amidst art marketization, urban consumerism, and neoliberalism, and a sustainable existentialist strategy employed by the *patuas* from time to time, even during the pandemic. Such resistance to marketizing tactics, whereby the market canonizes and fetishizes art as objects of curious, conspicuous consumption, is akin to Bill Ashcroft's postcolonial transformation that helps the minority rethink identity as a transformative political discourse, beyond an unproblematic transcendent absolute [89] (pp. 4–5):

> 'Today the means of representing cultural identity includes the whole range of plastic and visual arts, film and television and, crucially, strategies for consuming these products. Hence, trans-formation, which describes one way of viewing cultural identity, also describes the strategic process by which cultural identity is represented. By taking hold of the means of representation, colonized peoples throughout the world have appropriated and transformed those processes into culturally appropriate vehicles. It is this struggle over representation which articulates most clearly the material basis, the constructiveness and dialogic energy of the 'post-colonial imagination'.

The *patachitra* are not merely an outcome of an immutable, archaic traditional value system whose fossilized cultural capital may be exploited by the voyeuristic marketeers. Countering the dominant market (especially of foreign origin), *patuas* adopt radical, exploratory forms of *patachitra*. Beyond consuming the dominant cultural/technology/hegemony of the market, the *patuas* negotiate and critique the same.

The *patachitras* are not merely transformative in their expressions, but are also products that highlight sustainability. If the capitalist neocolonial market infrastructure exacerbates the socio-economic condition of the *patuas*, then the indigenous artistic traditions of *patachitra* across the modern platform and technology (YouTube, podcast, social media, Zoom talks, etc.) demonstrate postcolonial sustainable transformations. Such resistive transformations include discourse on various SDGs that impact grassroot democracy, social justice, ecological justice, climate action, and psycho-somatic well-being of market marginals. Next, we will discuss a few of their arts (before and during the pandemic) that are both traditional and disruptive, perhaps positively impacting SDG goals.

The goals of SDG 3 (good health and well-being) and SDG 4 (quality education) of the UN are to ensure healthy lives and well-being for all (at all ages) and quality education (here, awareness of COVID-19) [44]. This holistic attempt towards psycho-somatic well-being and good health is essential to sustainable development, especially for poor people such as the *patuas*. A third-world country such as India, which continues to face the devastating effects of COVID-19, must also fend itself from several other diseases and epidemics, malnutrition and maternity health issues. The poverty-related disease (PRD) of the marginal, vulnerable, and impoverished sections of India are primarily due to the lack of access to healthcare and unaffordable medical infrastructure [90]. COVID-19 created a huge income shock for the poor section of society, especially the agricultural laborers. 'Extreme poverty rose across India due to COVID-19 lockdown restrictions . . . . 44 million additional people fell into extreme poverty by July 2021' [90] (n.pg). Deepening the pre-existing inequalities between the 'haves' and the 'have-nots' of India, COVID-19 further negatively affected the dignity of life, health, and well-being of the poor.

Digital channels gained prominence during the lockdown. However, given the steep digital divide at the rural level of India [91], it was no surprise that the *patuas*, especially the senior artisans, either lacked access, opportunity, information, or usage of the medium [92]. Nevertheless, given the adoptable and resilient nature of the community, many of them shared their work through WhatsApp, which were uploaded to various social media platforms, such as Instagram and Facebook [92]. *Patachitra* artists, Swarna Chitrakar (~7 min YouTube clip; see Figure 2), Mamoni Chitrakar (~6 min YouTube clip; see Figure 3), Manu (aka Manoranjan), and Chitrakar (~3 min YouTube clip; see Figure 4), and their associated

songs to increase COVID-19 awareness highlighting virus detection procedure, self-hygiene, social distancing, general awareness, community surveillance, and information about government resources for the poor, became an internet sensation.

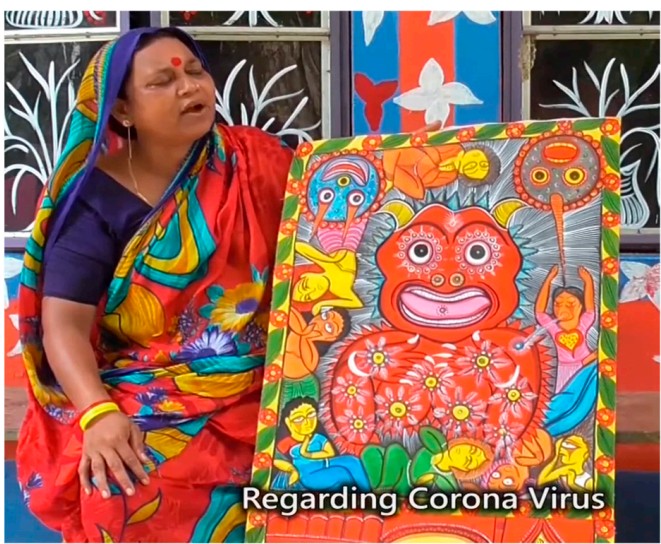

**Figure 2.** Swarna Chitrakar's art performance on COVID-19 awareness (https://www.youtube.com/watch?v=rNbQ5N59ZXE (accessed on 4 January 2023)).

The narrations also projected social issues of death and isolation of kith and kin, people living abroad, and their inability to travel back to India, request to the public for charity among the poor. Manu Chitrakar goes a step ahead by singing in Hindi, rather than Bengali, a language more accessible to the Indian masses. All three *patachitras* and the song exhibited positivity, hope, psychosomatic endurance, and importance of public well-being. Although traditionally invoking the Savior to protect all, the songs rationalized practical, scientific methods to avoid contact with the virus.

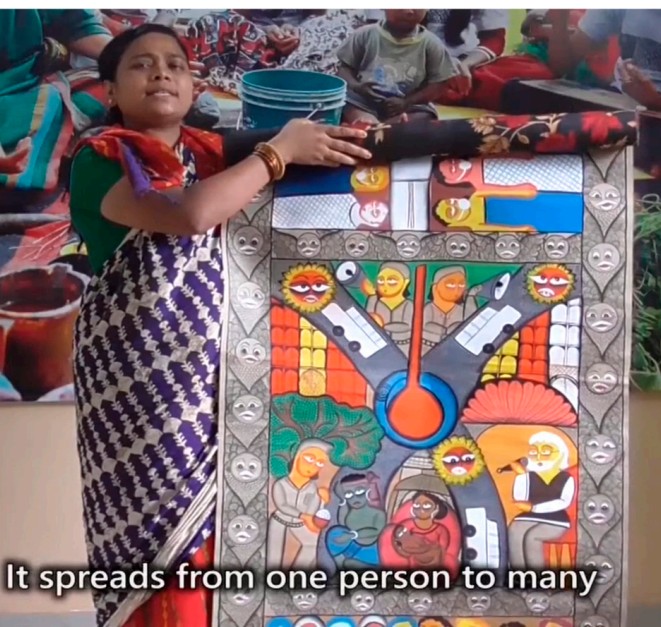

**Figure 3.** Mamoni Chitrakar's art performance on COVID-19 awareness (https://www.youtube.com/watch?v=iBx12Jgd5SA (accessed on 4 January 2023)).

Thus, the *patuas* made a transition to the digital world, re-strategizing the otherwise impenetrable social media infrastructure in rural India, to create sustainable awareness through their art form. Aided through the HIPAMS Project (Heritage-Sensitive Intellectual Property & Marketing Strategies)—a research project by *banglanatak dot com* and Coventry University—many *chitrakars* were able to reach the digital audience and expand their consumer base.

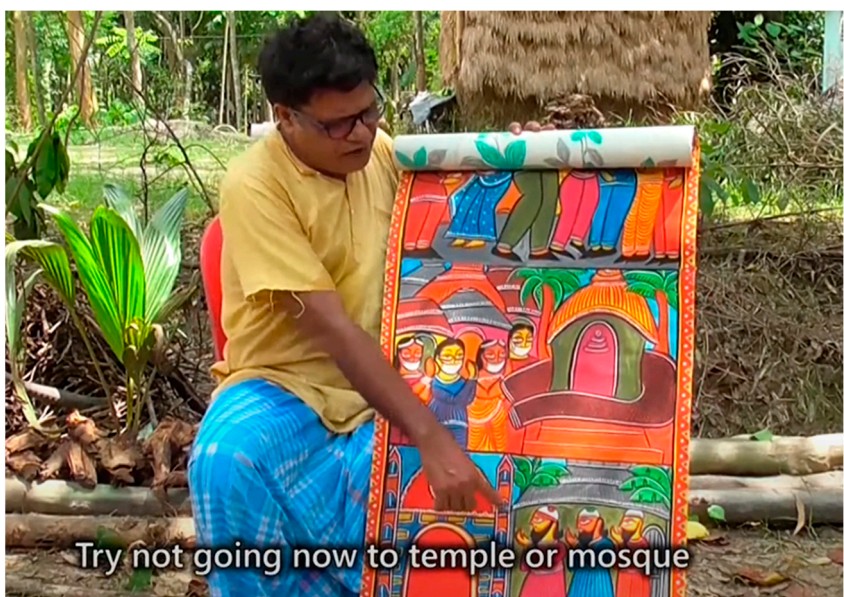

**Figure 4.** Manu Chitrakar's art performance on COVID-19 (https://www.youtube.com/watch?v=sjum5OU_bOE (accessed on 4 January 2023)).

The "HIPMUS Case Study" cites Sushama Chitrakar's acknowledgement of her digital de-subalternization:

'During the lockdown I have done many workshop programmes online and earned from it as well. Previously people would come to our village to buy products and attend workshop but now I can do all these online. We were unaware about online marketing before, HIPAMS has taught us to do that. We have learnt that business can also be done online by posting pictures of our product' [93].

Adopting the digital mode not only provided better exposure and prominence to the artisans, thereby consolidating their identity, but strategically created awareness among the public and Government about the craft-builder community and their condition during the pandemic [92]. The West Bengal Housing Infrastructure Development Corporation (HIDCO), a Government of West Bengal undertaking, later adopted the *patachitra* form to spread awareness about COVID-19 in Kolkata [94].

One may read into this strategy of the *patuas* as attempts at delivering SDG 3 and SDG 4 [44] during COVID-19. It is to be noted that *patachitras* have also been presumed as a visual narrative tool to generate community awareness about AIDS [95], environment-related issues [96], and gender equality [97]. Thus, *patachitras* could be visualized as elements that presume sustainability discourses and consolidate the identity of a marginalized local community. The penetration of the digital world and social media (access and usage), and the rethinking of the global/modern discourses of COVID-19 awareness through local sustainable practices has perhaps helped the community accentuate its art and identity. The community has thus attempted to avoid cultural erosions propagated by modernity/globalization and provided agency for self-representation. The community has also, through its *patachitras*, contributed, albeit unknowingly, to the SD discourses.

## 6. Discussion

There have been numerous arguments about how capitalism, neocolonization, and neo-liberalization are sustained, and how alternate indigenous economic systems are weakened, thereby impacting social justice, inclusivity, the production of sustainable counter knowledge, indigenous rights, the protection of health and well-being of the poor, and environmental/climate justice [98–104]. There is ample research about resistances to such hegemonic/anthropogenic market practices and indigenous negotiations of identity, consolidating the well-being of the community [105,106]. However, many have failed to see how such resistances to the market forces and the hegemony of modernist principles, which subsume the ontology of the marginal subjects, are cultural consumptions of sustainability. Although there have been attempts at analyzing sustainable consumption culture to probe consumer agency and sovereignty [107], such research lacks the critical focus in measuring the power dynamics between the neocolonial forces and the subaltern groups. Consumption from the margin, especially from the Global South, is an important critical topic of discussion in academia [108]. The sustainability angle in the subaltern consumer culture and the associated resistances are novel in this research. The appropriation of neoliberal and neocolonial forces and consumption of an alternate, sustainable, indigenous episteme have been the primary focus of our article. The environmental interests of *Bondas* (justice, democracy, revival of ethno-knowledge, resistance to neoliberalism) and the aesthetic practices of the *patuas* (appropriation of modern technological forces, community well-being, climate action) are consumptions of indigenous sustainable methods that advocate humanity, recognizes regenerative conservation, discourses the rights of indigenous masses, creates opportunity for indigenous agency, and are embedded with the everydayness of community well-being. The alternative consumption by two different subaltern rural communities of India fosters a socioecological matrix for a long-term conservation of community knowledge. It further contributes to the resistance register that forces the neoliberal market to reckon inclusivity, and articulates global sustainable goals through the local narratives of the margin.

To summarize our theoretical contributions, this article challenges the dominant discourse of sustainability and sees a counter narrative of sustainability consumption at the margin. By using a critical study, we highlight the uneven power relations within the communities examined, the society, and the market that underpin sustainability discourses and practices. These can help identify and address structural inequalities and injustices. The use of a critical lens assists us in probing the neoliberal/neocolonial market economy and its complex relationship palette with an ideology. We add to the existing qualitative research methods employed in critical sustainability studies and bridge them with consumer culture studies.

Finally, our research promotes reflexivity that will assist sustainability researchers and practitioners to rethink and review their privileges, assumptions, values, and biases.

In terms of practical implications, our findings may assist in developing an alternative sustainability discourse that runs counter to the standard 'normalized' market consumption practices. In addition, this research promotes equity and social justice by highlighting the power imbalances in the grand discourses of sustainability in the marketplace.

## 7. Conclusions

Critical sustainability studies from the Global South allude to a point of view that spotlights the effect of ecological and social sustainability issues according to the viewpoint of subalterns and marginalized communities in the developing world. This viewpoint recognizes that the ongoing worldwide arrangement of unreasonable asset distribution, which essentially helps well-off industrialized countries, frequently excessively affects the subalterns of the Global South. The critical sustainability studies from the Global South point of view requires a more even-handed circulation of assets, decision-making power, agency, and an acknowledgment of the diverse knowledge and experiences of communities in the Global South. This approach focuses on the requirements and encounters of the

marginal communities and perceives the significance of considering social, historical, and political settings while tending to sustainability issues. Through two important cases of cultural consumption and identity making of Indian marginal communities, *Bondas* and *Patachitrakars*, we illustrate the impact of traditional sustainable knowledge consumption and how it shapes identity as different from that of the neoliberal market order. We project how in recreating the voices of their subaltern community and expressing their socio-cultural value system, the *Bondas* and the *Patachitrakars* seize agency and subvert the dominant global market logic.

## 8. Future Research and Limitations

Any future research that would further contribute to the critical study of sustainability discourses of the Indigenous consumer cultures may analyze various other cases from the Global South. Stories and performances of power/market-force negotiations from the decolonizing spaces would surely add to the fledging 'critical sustainability research through consumer culture', an area that this journal Special Issue calls for. Moreover, future research may include an intersectional approach between critical sustainability studies, gender/race/class consumption. In this paper, we have largely emphasized the perspective of decolonization, a sustainable strategy for the marginalized. Future research may further extend the postcolonial/decolonial agenda in sustainable consumption of marginal communities from the Global South. In addition, future research may also probe the alternate green consumption and claims of green justice of marginal communities. Future research may work closely with subaltern communities to understand their perspectives and experiences in a culturally aware and regionally grounded way through participatory research. This was our limitation due to lack of funding and the COVID-19 pandemic that put sanctions on the general public in India to have in-person engagement with members of the marginal community. To make any grand claim that as researchers we have been able to conclusively validate the decolonizing, sustainable process of the *Bonda's* agricultural practices and the *patua*'s identity articulation thorough re-negotiation of their art, would be an overstatement. This new research paradigm that prognosticates alternate equitable futures, initiated by our article, has a long way to go.

Our straddling of consumer culture studies and critical sustainability study intends to provide new insights and perspectives on sustainability consumption of the subalterns at the margin. However, the research is not free from its own limitations. We focus on specific case studies and, as with any qualitative research, these limit the generalizability of our findings.

**Author Contributions:** Conceptualization, A.D.; methodology, A.D.; formal analysis, A.D. and P.A.A.; investigation, A.D. and P.A.A.; resources, A.D. and P.A.A.; writing—original draft preparation, A.D.; writing—review and editing, P.A.A. All authors have read and agreed to the published version of the manuscript.

**Funding:** This research received no external funding.

**Institutional Review Board Statement:** Not applicable.

**Informed Consent Statement:** Not applicable.

**Data Availability Statement:** Not applicable.

**Conflicts of Interest:** The authors declare no conflict of interest.

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
