# Peer review of "Consumption Culture and Critical Sustainability Discourses: Voices from the Global South"

_sustainability, doi:10.3390/su15097719_

Round 1
Reviewer 1 Report (Previous Reviewer 1)
All comments are well answered, the paper is now improved.
Author Response
We thank all the Reviewers for their patient readings and insightful comments. We have incorporated all the changes, as advised.

Reviewer 2 Report (New Reviewer)
The manuscript is clear and relevant to the field. It is well-structured. In my opinion, the paper will attract the readership.
The manuscript discusses the important issue of consumption culture and sustainability and analyzes communities in India. The topicality of the issue is explained.
The manuscript properly demonstrates its case. Research objectives and research questions are well-defined and original. The results are interpreted appropriately.
Future research directions are suggested, however, the limitations of this research are not discussed. Also, the theoretical and practical implications of the study could be presented.
There are repetitive sentences in Lines 83-84 and 95-96.
Author Response
We thank all the Reviewers for their patient readings and insightful comments. We have incorporated all the changes, as advised.

Reviewer 3 Report (New Reviewer)
I found this a highly engaging paper. It is positioned, polemic and impactful. It is well embedded in relevant literature and debates and carves out a convincing case for, and example of, research in the field of subaltern sustainable consumption cultures and practices. I have included my comments, which it makes no sense to repeat here, as they are embedded in the paper in the context where they apply. On the whole my comments relate to highly particular matters, including issues of grammar, clarity of meaning, and some points made that I feel are not warranted or sufficiently justified in one way or another. At a more substantial level I do find the methodology section lacking. On the one had there is very little to illuminate any of the substantive processes you undertook to subject this data to analysis. In this sense the methodology section appears under-developed. However, at the same time, the reference you make to have been able to validate your data analysis and your claim to have a methodology able to offer insight into the unconscious dimensions of what you are examining via this data appear over-stated. I can's see that either of these claims are supportable. I would suggest you moderate your claims here and expand a little on how they undertook the data selection and analysis, and some commentary on the limitations of the data. A substantive paragraph should suffice. Please address the issues raised and I would consider this paper a valuable contribution to the field

Author Response
We thank all the Reviewers for their patient readings and insightful comments. We have incorporated all the changes, as advised.

This manuscript is a resubmission of an earlier submission. The following is a list of the peer review reports and author responses from that submission.
Round 1
Reviewer 1 Report
A summary: This research critically examines sustainability discourse in the Global South through a sociological critique of anti-sustainable consumption. The study highlights the resistance to market forces and hegemonic principles through sustainable consumption culture in subaltern communities.
General concept comments
- the overall structure of the paper is good however it is required a redefinition,
- The results are very satisfying however the methodology is not clear.
- The motivation behind this work is not clearly stated.
- The use of visual aids images and videos are very clear and disserve a praise.
- The practical implications of the findings is clearly highlighted
Specific comments:
Major points
Here are some suggestions to improve the work:
Include a section outlining the research technique.
Additionally, the purpose of this work and its final objective must be made clear.
To make the results more applicable across the Global South, think about incorporating a larger variety of localities and areas.
To further enliven the analysis, think about combining viewpoints from different fields, such as political science, economics, or environmental science.
Work closely with subaltern communities to understand their perspectives and experiences in a culturally aware and regionally grounded way by engaging in more participatory research.
Minor points
Include a section of relevant work that addresses the limits and gaps in the existing literature or acknowledge them in the introduction.
Although the findings are crucial, the discussion of the discussion conclusion and future work is weak. As a result, we recommend breaking the sections into two parts: discussion and future work, followed by conclusion.
If there are some statistics it is recommended to add it in the paper,
Table 1 and 2 are not addressed in the text: it should be addressed.
Reconsider to redesign the table 1 in a flow diagram to better describe the steps
Reconsider just discussing the table 2 because it states only advantages it can be discussed in a paragraph form
Reviewer 2 Report
Improve writing to concretize ideas and maintain interest.
I do not identify the methodology, could you explain how this review was carried out please.
Separate discussion from conclusions, and mention limitations and future directions separately
Update references older than 20 years and review the style of all references, that the links correspond and add the numbering of the pages of the books in the references.